# Prognostic accuracy of the CMPMIT-ICD-10, APACHE II, SOFA, ISS, and AIS for in-hospital death among patients with traumatic hemorrhagic shock

Jie Cai[1,2,3], Gang Zhou[1,2], Feifei Jin[2,4,5], Haiyan Xue[1,2], Shu Li[1,2], Chun Fu[1,2], Zhenzhou Wang[1,2], Wei Huang[2,4,5], Tianbing Wang[2,4,5], Fengxue Zhu[1,2]*, Xiujuan Zhao[1,2]*

1 Department of Critical Care Medicine, Peking University People's Hospital, Beijing, China, 2 Trauma Medicine Center, Peking University People's Hospital, Beijing, China, 3 Department of Critical Care Medicine, Shenzhen Nanshan People's Hospital,Shenzhen, China, 4 Key Laboratory of Trauma and Neural Regeneration (Peking University), Ministry of Education, Beijing, China, 5 National Center for Trauma Medicine of China, Beijing, China

☯ These authors contributed equally to this work.
* fengxue_zhu@126.com (FZ); zxj_0515@163.com (XZ);

## Abstract

The China Mortality Prediction Model in Trauma, which is based on the International Classification of Disease Disorders (ICD)-10-CM lexicon (CMPMIT-ICD-10),is a new trauma scoring system. Our objective was to compare the prognostic performance of the CMPMIT-ICD10 with that of the Acute Physiology and Chronic Health Evaluation II (APACHEII), Sequential Organ Failure Assessment (SOFA), Injury Severity Score (ISS), and Abbreviated Injury Scale (AIS) for in-hospital mortality in patients with traumatic hemorrhagic shock(THS).This retrospective observational cohort study was conducted at a tertiary teaching hospital from May 1, 2013, to May 31, 2023. The area under the receiver operating characteristic curve (AUC), sensitivity, specificity, accuracy, and associations with outcomes of theCMPMIT-ICD-10, APACHE II, SOFA, ISS, and AIS scores for the prediction of in-hospital death were assessed. A total of 420 patients with THS were included. Forty-one (9.8%) patients died during hospitalization. For the prediction of in-hospital death, the CMPMIT-ICD-10 (0.8757) and APACHE II(0.8709) had greater AUCs compared with the AIS (0.6243), SOFA (0.7669), and ISS (0.6601). With the best cut-off value of 59.5, the CMPMIT-ICD10 had a highest sensitivity (85.4%) and good specificity(79.9%) and overall accuracy (80.4%). The CMPMIT-ICD10 (OR 1.057, 95% CI 1.028–1.087, $p < 0.001$) and APACHE II (OR 1.125, 95% CI 1.045–1.211, $p = 0.002$) were independently associated with in-hospital death. Comparable to the APACHE II but significantly better than the SOFA, ISS, and AIS, the CMPMIT-ICD-10 performed well in predicting the short-term mortality of patients with THS. These findings suggest that the CMPMIT-ICD-10 may have superior utility for predicting short-term death in THS patients.

**Data availability statement:** All relevant data are within the manuscript and its Supporting Information files.

**Funding:** This study was supported by National Natural Science Foundation of China in the form of a grant awarded to FZ (81971808), Beijing Natural Science Foundation in the form of a grant awarded to FZ (7222199), Capital Health Research and Development of Special in the form of a grant awarded to FZ (2018-2-4082) and Peking University People's Hospital Scientific Research Development Funds in the form of a grant awarded to FZ (RDGS2023-07). The specific roles of this author are articulated in the 'author contributions' section. The funders had no role in study design, data collection and analysis, decision to publish, or preparation of the manuscript.'.

**Competing interests:** The authors have declared that no competing interests exist.

## Introduction

Severe trauma remains the leading cause of mortality and disability in young adults [1]. Deaths from severe trauma account for 10% of all deaths [2], and hemorrhagic shock related to trauma is the main cause of early in-hospital death, accounting for 17.1% to 50% of all trauma-related deaths [3–5].Trauma scoring systems play a crucial role in the quality of trauma care, as they are a means of horizontal and vertical comparisons of baseline trauma care quality and objective measurement for clinicians to assess the severity of injuries, predict patient prognosis, and optimize resource allocation.

The two main types of scoring systems currently used in the assessment of trauma patients include physiological scoring systems based on physiological indicators, such as the Acute Physiology and Chronic Health Evaluation (APACHEII), and the Sequential Organ Failure Assessment (SOFA), and anatomical scoring systems based on the location and extent of injuries, such as the Abbreviated Injury Scale (AIS), Injury Severity Score (ISS), New Injury Severity Score (NISS), and International Classification of Diseases Tenth Revision(ICD-10)-based Injury Severity Score (ICISS). Each type of trauma scoring system has advantages and disadvantages. For example, physiological scoring systems are advantageous because they are objective, easy to use, and have readily available indicators; however, these scoring systems are easily affected by treatment and do not reflect the anatomical damage caused by trauma, thus failing to comprehensively reflect the severity of injuries [6]. Although anatomical scoring systems can accurately assess the severity of injuries and predict patient prognosis, they must be used by highly skilled professionals and cannot dynamically assess changes in the patient's condition [7].

The China Mortality Prediction Model in Trauma based on the ICD-10-CM lexicon (CMPMIT-ICD-10) is a trauma-related mortality risk prediction model that was constructed by our team on the basis of the International Classification of Diseases, Tenth Revision (ICD-10) coding system and a database of trauma patients in China and accounts for the impact of previous comorbidities and post traumatic physiological responses on mortality risk [8]. This model performed well on the development and internal validation datasets. This prediction model not only improves clinicians' awareness of the risk of mortality among trauma patients but can also be used to adjust for differences in patient case mix and disease severity across hospitals.

Current prognostic scores for trauma may not be adequate for patients with traumatic hemorrhagic shock (THS), and the effectiveness and accuracy of the CMPMIT-ICD-10 score for assessing the risk of mortality in patients with THS have not been fully validated or compared. The aim of this study was to assess and compare the prognostic performance of the CMPMIT-ICD-10 score, APACHE II score, SOFA score, ISS and AIS score for in-hospital mortality in patients with THS.

## Materials and methods

### Study design

This retrospective observational cohort study was conducted at the Peking University People's Hospital, a tertiary teaching hospital of Peking University, China. The Peking

University People's Hospital Medical Ethics Board approved this study (2020PHB258−01). Informed consent for participation in this study was waived because no treatment interventions were mandated, and no protected health information was collected or analyzed. This study was conducted in accordance with the Strengthening the Reporting of Observational Studies in Epidemiology (STROBE) statement and the Declaration of Helsinki [9].

## Patients

All adult patients (≥18 years old) who were consecutively diagnosed with THS and who stayed in the ICU for more than 24 h between May 1, 2013, and May 31, 2023, were included. The diagnostic criteria for THS at admission were as follows (meeting a,b,c or a,b, and d): a, obvious bleeding caused by trauma (estimated blood loss volume greater than 1200 ml); b, a hemoglobin concentration<100 g/L or >30 g/L and lower than that before trauma;c, a systolic blood pressure <90 mmHg (or shock index [heart rate/systolic blood pressure] >1) for three consecutive measurements; and/or d, a serum lactate concentration>2 mmol/L [10]. Patients who were pregnant or lactating, who were suffering from shock due to other causes, who were classified as non-traumatic, who were expected to die within 24 h of admission because of fatal trauma, or who had missing data were excluded. All patients who were evaluated, diagnosed, and treated for THS by the same team of critical care specialists followed the same treatment protocol derived from available recommendations [11–12].

## Data collection

Data were collected consecutively and retrospectively. All the data were obtained from the Trauma-Specific Database, which is a real-world clinical database composed of data from more than 23,000 trauma patients registered between 2012 and May 2023. To ensure the highest data integrity and accuracy, we implemented a multilayered validation process. This included automated validation rules, logic and consistency checks, and regular audits and reconciliation. This rigorous approach was crucial for maintaining the database's reliability and accuracy for both clinical decision-making and research analysis.Trained doctors and research nurses input the data. They were unaware of the study protocol and did not participate in the management or care of the patients. Ten percent of the sample data were randomly selected to assess the quality of the data. We started accessing the data on September 25, 2022.

Patient demographics, comorbidities, causes of trauma, main bleeding site, and other laboratory and clinical variables required for the assessments of ISS, AIS scores, APACHE II scores, SOFA scores, and CMPMIT-ICD-10scores were collected. The worst value of each score within 24 h after ICU admission was used in this study.

The China Mortality Prediction Model in Trauma based on the ICD-10-CM lexicon (CMPMIT-ICD-10) is a new trauma scoring system based on data from the Beijing Red Cross Emergency Center [8]. This scoring system accounted for sex, age, partial new injury codes, partial comorbidities, traumatic shock, and state of consciousness.The scores of this scale are shown in Table 1, and the ICD-10codes are listed in Table S1 in S1 File.The risk of death was graded as follows: 0–47,extremely low risk (risk of mortality is <10%); 48–60,low risk (risk of mortality is 11–30%); 61–73,medium risk (risk of mortality is 31–60%); 74–90,high risk (risk of mortality is 61–90%); and >90,extremely high risk (risk of mortality is >90%). The total scores ranged from 0 to 232. To facilitate the use of CMPMIT-ICD-10, the National Center for Trauma Medicine of China developed a WeChat Mini Program.

Acute respiratory distress syndrome was defined according to the Berlin definition [13]. Acute kidney injury was defined as a serum creatinine increase of ≥0.3 mg/dL (≥26.5 μmol/L) within 48 h or a urine output volume <0.5 mL/(kg·h) for 6 h [14]. Acute liver injury was detected and confirmed by liver biochemical blood tests in addition to a lack of history of acute or chronic hepatitis or liver cirrhosis. The definitions for acute liver injury include one of the following thresholds: i) ≥5 × upper limit of normal(ULN) elevation in alanine aminotransferase, ii) ≥2×ULN elevation in alkaline phosphatase, or iii) ≥3×ULN elevation in alanine aminotransferase and simultaneous elevation of total bilirubin concentration exceeding 2× ULN [15]. Myocardial injury was defined as a cardiac troponin I(cTNI) concentration above the 99th percentile upper reference limit. Injury was considered acute if there was an increase and/or decrease in cTNI values [16]. cTNI levels were assessed via a high-sensitivity troponin

**Table 1. CMPMIT-ICD-10 scores.**

| Variables | Categories | Scores |
|---|---|---|
| Sex | Male | 3 |
| Age | ≤40 | 0 |
| | 41-50 | 4 |
| | 51-60 | 10 |
| | 61-70 | 11 |
| | 71-80 | 18 |
| | 81-90 | 23 |
| | ≥91 | 27 |
| A2 | Yes | 5 |
| A3 | Yes | 7 |
| A4 | Yes | 16 |
| A5 | Yes | 17 |
| D3 | Yes | 2 |
| E2 | Yes | 10 |
| E3 | Yes | 22 |
| F2 | Yes | 11 |
| G3 | Yes | 4 |
| Traumatic shock | Yes | 28 |
| Coma | Yes | 29 |
| Myocardial infarction | Yes | 6 |
| Congestive heart failure | Yes | 16 |
| Chronic renal failure | Yes | 16 |
| Cerebrovascular diseases | Yes | 3 |
| Peptic ulcer | Yes | 10 |

A2…G2, G3 are new region-severity codes. See S1 Table for details.

I assay on a DxI800 (Beckman Coulter, Brea, CA, USA), wherein the 99th percentile for this test was 0.034 ng/mL.Trauma-induced coagulopathy was defined as the presence of significant coagulopathy upon arrival, a prothrombin time>18 seconds, an activated partial thromboplastin time>60 seconds, or a thrombin time>15 seconds (1.5 times the normal value). According to the British National Blood Transfusion Service and the American College of Pathologists,coagulopathy is characterized by the need for blood product replacement therapy in the presence of active or impending hemorrhage [17–19]. If such organ dysfunction occurred, effective treatment was administered according to the corresponding guidelines or standards of care.

## Outcomes

The primary outcome was in-hospital death. The patients' discharge data were retrieved from the trauma-specific data-base at the hospital and screened for in-hospital death.

## Statistical analysis

Using power analysis and sample size (PASS)15.0 for sample size calculation, we calculated the required sample size on the basis of the area under the ROC curve of the model(AUC0 = 0.5 for the null hypothesis and AUC1 = 0.65 for the alternative hypothesis). The ratio of nonsurvival to survival was 1:9 (α = 0.05; power = 0.8). A total of 330 participants (no fewer than 33 nonsurviving patients) were included. Approximately 20% of the participants dropped out; thus, a total of 396 participants were included.

Continuous variables are presented as the mean (standard deviation, SD) or median (interquartile range, IQR), as appropriate, and categorical variables are presented as counts (percentages). Student's t test, the nonparametric Mann–Whitney $U$ test, and the Pearson $\chi 2$ test were used for comparisons of variables. The missing data rate of all variables was < 5%, missing continuous variables were inferred as the median of nonmissing values, and missing categorical covariates were inferred as the most frequent categorical values.

The discriminative power of the ISS, AIS, APACHEII score, SOFA score, and CMPMIT-ICD-10 score was first assessed using the area under the receiver operating characteristic curve (AUC) for the prediction of death. The cutoff value of each score was determined using the Youden index. The sensitivity, specificity, positive predictive value (PPV), negative predictive value (NPV), and accuracy of these five scores at the corresponding cutoff values were analyzed and compared. The DeLong test was used to calculate the 95% CI of the AUC, and the McNemar test was used to compare the AUCs between the scores.

Next, we examined the predictive values of the ISS, AIS score, APACHE II score, SOFA score, and CMPMIT-ICD-10 score using logistic regression analyses. Moreover, we conducted a subgroup analysis on the basis of the presence of multiple organ dysfunction syndrome (MODS) and length of ICU stay and a sensitivity analysis on the basis of mechanical ventilation. MODS was defined as the occurrence of dysfunction of two or more organs within one week of traumatic hemorrhagic shock, with a SOFA score ≥ 4 points [20].

Statistical analyses were conducted using SPSS (version 22.0) and R (version R4.3.2). Statistical thresholds were adjusted for multiple comparisons using the Bonferroni correction.

## Results

Among the 505 patients with THS who were treated at this hospital, 15 patients who were pregnant, 18 patients who died within 24 h of admission, and 52 patients whose data were missing were excluded. Thus, 420 patients were included in the analysis (Fig 1). The patients' demographic and clinical indicators are presented in S2 Table . Forty-one (9.8%) patients died, and 379 (90.2%) patients survived during hospitalization. The median patient age was 60 (48–78) years, and 55.5% were male.

Comparisons of clinical characteristics between the survival and nonsurvival groups are shown in Table 2. The CMPMIT-ICD-10 score (85 [69–90] versus 51 [45–57], $p < 0.001$), APACHE II score (27[23–34.5] vs. 17[13-20], $p < 0.001$), SOFA score (9[5–12] vs. 4[2–6], $p<0.001$), ISS (26[17–36] vs. 19[14–29], $p = 0.001$) and AIS score (10[7–18] vs. 10[7–14], $p = 0.009$) were significantly greater in the nonsurvival group compared with the survival group.

For the prediction of in-hospital death, the CMPMIT-ICD-10 score had the highest AUC (0.8757, 95% CI 0.8194–0.932), whereas the AIS score had the lowest AUC (0.6243, 95% CI 0.5295–0.7191) (Table 3, Fig 2A). A significant difference was noted between the CMPMIT-ICD-10 score and the SOFA score and between the ISS and AIS score ($p<0.05$) (Fig 2B). The second highest AUC was for the APACHE II score (0.8709, 95% CI 0.8152–0.9267), and the second lowest was for the ISS (0.6601, 95% CI 0.5657–0.7546) (Table 3, Fig 2A). A significant difference was noted between the APACHE II score and the SOFA score and between the ISS and AIS ($p<0.05$) (Fig 2B).

The sensitivity, specificity, PPV, NPV, and accuracy of these five scores with the best cutoff values are presented in Table 3 and Fig 3. For the prediction of in-hospital mortality, the CMPMIT-ICD-10 score, with the best cutoff value of 59.5, had the highest sensitivity (0.854), and the APACHE II score, with the best cutoff value of 22.5, had the highest specificity (0.839). The APACHE II score had the highest accuracy (0.833), followed by the CMPMIT-ICD-10 score (0.804).

Logistic regression analyses of these five scores are presented in Table 4. According to the univariate analyses, all five scores were significantly associated with in-hospital death, and multivariate logistic regression analyses revealed that the CMPMIT-ICD-10 score (OR 1.057, 95% CI 1.028–1.087, $p<0.001$) and the APACHE II score (OR 1.125, 95% CI 1.045–1.211, $p=0.002$) were independently associated with in-hospital death.

Sensitivity and subgroup analyses were performed to evaluate the robustness of our primary results.

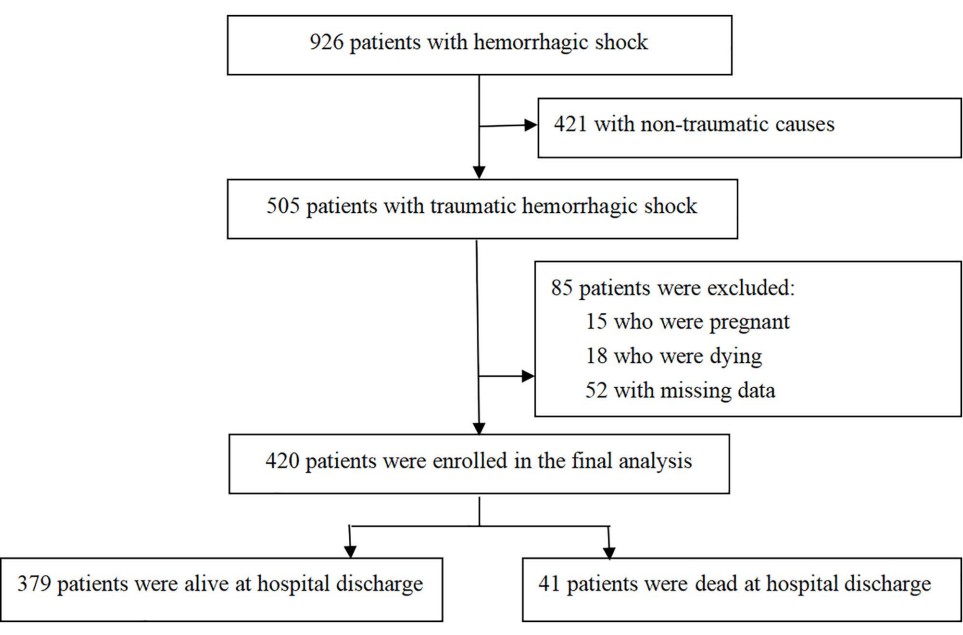

**Fig 1. Flowchart.**

First, we excluded non-mechanically ventilated patients and evaluated the relationships between these five scores and in-hospital death among mechanically ventilated patients. In mechanically ventilated patients, the CMPMIT-ICD-10 score had the highest AUC (0.8663, 95% CI 0.8101–0.9225), the highest sensitivity (0.800), good specificity (0.814) and good accuracy (0.812) for predicting in-hospital death (S3-4 Tables, S1-2 Figs in S1 File), and the CMPMIT-ICD-10 score (OR 1.050, 95% CI 1.021–1.080, $p < 0.001$) was independently associated with in-hospital death in mechanically ventilated patients (Table 4).

Second, we divided all the patients into a MODS group and a non-MODS group. The results revealed that the CMPMIT-ICD-10 score had the second highest AUC (0.8706, 95% CI 0.7953–0.9459), with the highest sensitivity (0.867), slightly worse specificity (0.783) and good accuracy (0.800) for predicting in-hospital death in MODS patients (S5-6 Tables, S3-4 Figs in S1 File), and the CMPMIT-ICD-10 score (OR 1.060, 95% CI 1.021–1.100, $p = 0.002$) was an independent risk factor for in-hospital death in MODS patients. In non-MODS patients, the CMPMIT-ICD-10 score had the highest AUC (0.8708, 95% CI 0.7780–0.9637), with greater sensitivity (0.818), good specificity (0.842) and accuracy (0.841) for predicting in-hospital death (S5-6 Tables, S3-4 Figs in S1 File), and the CMPMIT-ICD-10 score (OR 1.099, 95% CI 1.039–1.162, $p < 0.001$) was an independent risk factor for in-hospital death in non-MODS patients (Table 4).

Third, according to the ICU length of stay, we divided all patients into two groups: those whose ICU length of stay was < 7 days and those whose ICU length of stay was ≥ 7 days. For patients with longer (≥7 days) ICU stays, the CMPMIT-ICD-10 score had the highest AUC (0.8305, 95% CI 0.7544–0.9066), a slightly worse sensitivity (0.762), and good specificity (0.783) and accuracy (0.781) for predicting in-hospital death (S7-8 Tables, S5-6 Figs in S1 File).The CMPMIT-ICD-10 score (OR 1.040, 95% CI 1.002–1.078;$p = 0.036$) was independently associated with in-hospital death (Table 4). However, in patients with shorter (<7 days) ICU stays, the CMPMIT-ICD-10 score had a greater AUC (0.9195, 95% CI 0.8327–1.000), the second highest sensitivity (0.900), good specificity (0.875) and accuracy (0.878) for predicting in-hospital death (S7-8 Tables, S5-6 Figs in S1 File), and the CMPMIT-ICD-10 score was not associated with in-hospital death (Table 4).

**Table 2. Comparison of characteristics between nonsurviving and surviving patients with traumatic hemorrhagic shock.**

| Characteristics | Non-Survival group n=41 | Survival group n=379 | p Value |
|---|---|---|---|
| Male, n(%) | 27 (65.9) | 206(54.4) | 0.159 |
| Age, years, median (IQR) | 60 (47.0–77.0) | 62(50.5–81.5) | 0.352 |
| Comorbidities, n(%) | | | |
| Stroke | 6(14.6) | 27 (7.1) | 0.164 |
| Coronary heart disease | 3 (7.3) | 19(5.0) | 0.795 |
| Hypertension | 18 (43.9) | 108(28.5) | 0.041 |
| Chronic obstructive pulmonary disease | 2(4.9) | 12(3.2) | 0.903 |
| Diabetes mellitus | 7(17.1) | 46(12.1) | 0.366 |
| Chronic kidney disease | 1(2.4) | 5(1.3) | 0.566 |
| Cancer | 2(4.9) | 27(7.1) | 0.830 |
| Causes of trauma, n(%) | | | |
| Falling from height | 9(22.0) | 59(15.6) | 0.292 |
| Road traffic accident | 13(31.7) | 155(40.9) | 0.254 |
| Falling from a standing position | 16(39.0) | 134(5.4) | 0.641 |
| Others (crush, stab, animal bite) | 3(7.3) | 31(8.2) | 0.848 |
| Main bleeding site, n(%) | | | |
| Thoracic | 9(22.0) | 70 (18.5) | 0.383 |
| Abdominal | 4(9.8) | 48(12.7) | 0.561 |
| Pelvic | 8(19.5) | 48(12.7) | 0.112 |
| Limbs | 7(17.1) | 158(41.7) | 0.002 |
| Others (blood vessels, skin, and soft tissue) | 13(31.7) | 55(14.5) | 0.005 |
| Mean arterial pressure, mmHg, median (IQR) | 69.3(53.2–80.5) | 72(61.7–86.0) | 0.096 |
| Heart rate, beats/min, median (IQR) | 123.0(103.0–138.8) | 106.0(89.0–121.0) | 0.001 |
| Laboratory test | | | |
| Leukocyte count, cells×$10^9$/L, median (IQR) | 9.8(7.1–17.5) | 10.7(8.2–15.7) | 0.739 |
| Hemoglobin, g/L, mean±SD | 89.9±35.3 | 96.2±22.6 | 0.117 |
| Platelet count, cells×$10^9$/L, median (IQR) | 101.5(27.8–192.5) | 138.0(89.0–187.5) | 0.007 |
| Serum creatinine, μmol/L, median (IQR) | 103.0(69.0–136.5) | 70.0(57.0–97.0) | <0.001 |
| Total bilirubin, μmol/L, median (IQR) | 13.4(8.4–23.7) | 15.5(10.8–22.5) | 0.655 |
| Prothrombin time, sec, median (IQR) | 14.9(12.3–18.9) | 12.9(11.8–14.9) | 0.001 |
| Fibrinogen, mg/dL, median (IQR) | 159.0(76.5–283.1) | 235.0(149.0–333.0) | 0.002 |
| pO$_2$/FiO$_2$ ratio, mmHg, median (IQR) | 282.5(153.5–367.9) | 320.0(236.7–388.0) | 0.035 |
| Serum lactate, mmol/L, median (IQR) | 5.8(2.1–11.3) | 2.5(1.5–3.6) | <0.001 |
| Serum Procalcitonin, ng/mL, median (IQR) | 7.6(0.6–37.2) | 1.5(0.4–4.3) | 0.051 |
| Cardiac troponin I, pg/mL, median (IQR) | 280.5(308.4–1008.6) | 43.7(10.3–229.5) | 0.002 |
| B-type natriuretic peptide, pg/mL, median (IQR) | 110.5(52.0–515.0) | 59.0(25.3–164.5) | 0.008 |
| Organ dysfunction, n(%) | | | |
| Acute respiratory distress syndrome | 18(43.9) | 92(24.3) | 0.007 |
| Acute kidney injury | 27(65.9) | 89(23.5) | <0.001 |
| Acute myocardial injury | 33(80.5) | 174(45.9) | <0.001 |
| Trauma-induced coagulopathy | 19(46.3) | 42(11.1) | <0.001 |
| Acute liver injury | 18(43.9) | 105(27.7) | 0.030 |
| Scores, median (IQR) | | | |
| CMPMIT-ICD-10 | 85.0(69.0–90.0) | 51.0(45.0–57.0) | <0.001 |

*(Continued)*

**Table 2.** (Continued)

| Characteristics | Non-Survival group<br>n = 41 | Survival group<br>n = 379 | p Value |
|---|---|---|---|
| APACHEII | 27.0(23.0–34.5) | 17.0(13.0–20.0) | <0.001 |
| SOFA | 9.0(5.0–12.0) | 4.0(2.0–6.0) | <0.001 |
| ISS | 26.0(17.0–36.0) | 19.0(14.0–29.0) | 0.001 |
| AIS | 10.0(7.0–18.0) | 10.0(7.0–14.0) | 0.009 |

IQR, Interquartile range; pO₂/FiO2 ratio,Arterial partial pressure of oxygen to fraction of inspired oxygen ratio; CMPMIT-ICD-10, China Mortality Prediction Model in Trauma based on the ICD-10-CM lexicon; APACHEII, Acute Physiology and Chronic Health Evaluation; SOFA, Sequential Organ Failure Assessment; ISS, Injury Severity Score; AIS, Abbreviated Injury Scale.

**Table 3.** AUCs and best cutoff values, sensitivity, specificity, PPV, NPV, and accuracy of the five scores.

| Variables | AUC(95% CI) | p value | Cutoff value | Sensitivity(%) | Specificity(%) | PPV(%) | NPV(%) | Accuracy(%) |
|---|---|---|---|---|---|---|---|---|
| CMPMIT-ICD-10 | **0.8757(0.8194–0.9320)** | 0.029 | 59.5 | **85.4** | 79.9 | 31.5 | **98.1** | 80.4 |
| APACHEII | 0.8709(0.8152–0.9267) | 0.029 | 22.5 | 78.0 | **83.9** | 34.4 | 97.2 | **83.3** |
| SOFA | 0.7669(0.6812–0.8526) | 0.043 | 7.5 | 63.4 | 80.5 | 26.0 | 95.3 | 78.8 |
| ISS | 0.6601(0.5657–0.7546) | 0.048 | 23.5 | 70.7 | 61.7 | 16.7 | 95.1 | 62.6 |
| AIS | 0.6243(0.5295–0.7191) | 0.048 | 12.5 | 56.1 | 66.8 | 15.4 | 93.4 | 65.7 |

AUC, area under the receiver operating characteristic curve; PPV, positive predictive value; NPV, negative predictive value

## Discussion

This study revealed an in-hospital mortality rate of 9.8% for THS patients, whereas a multicenter study from Europe and the USA revealed a mortality rate of 36.5% for THS patients with severe injuries [21]. Accurate early prediction of the risk of death may aid in triage decisions and treatment strategies. This study is the first to compare the prognostic performance of the new trauma score (CMPMIT-ICD-10) and four broader critical illness or trauma scores (the APACHE II score, SOFA score, ISS, and AIS score) for mortality in critically ill patients diagnosed with THS. We found that compared with the SOFA score, ISS, and AIS score, the CMPMIT-ICD-10 and APACHE II scores had excellent predictive value for the risk of death in THS patients. The CMPMIT-ICD-10 and APACHE II score were identified as independent risk factors for in-hospital death in patients with THS. The CMPMIT-ICD-10 may be an important tool for predicting the risk of death, triage decisions, and treatment strategies in patients with THS.

### Advantages and disadvantages of several scores in trauma patients and the proposedCMPMIT-ICD-10 score

To date, several scoring systems have been developed in an attempt to predict outcomes in trauma patients, including anatomically based scoring systems, such as the ISS [22] and the NISS [23], which were derived from the AIS [24]; ICD-based scoring systems, such as the ICISS [25] and the TMPM-ICD-10 [26]; and general ICU scoring systems, such as the APACHE II [27] and the SOFA [28]. Unfortunately, all of these scores have limitations. The AIS score and ISS do not account for physiologic variables and require coders to be trained, which can be time consuming. In addition, accurate coding is often not possible in the initial stages of trauma. The APACHE II score and SOFA score are not indicators of anatomical injuries caused by trauma; thus, they cannot fully represent the severity of injury. Furthermore, these two scores were the worst in the first 24 hours after admission to the ICU and are susceptible to initial resuscitation and the level of care in the ICU. Because the current scoring systems cannot predict outcomes promptly and accurately, our team

A

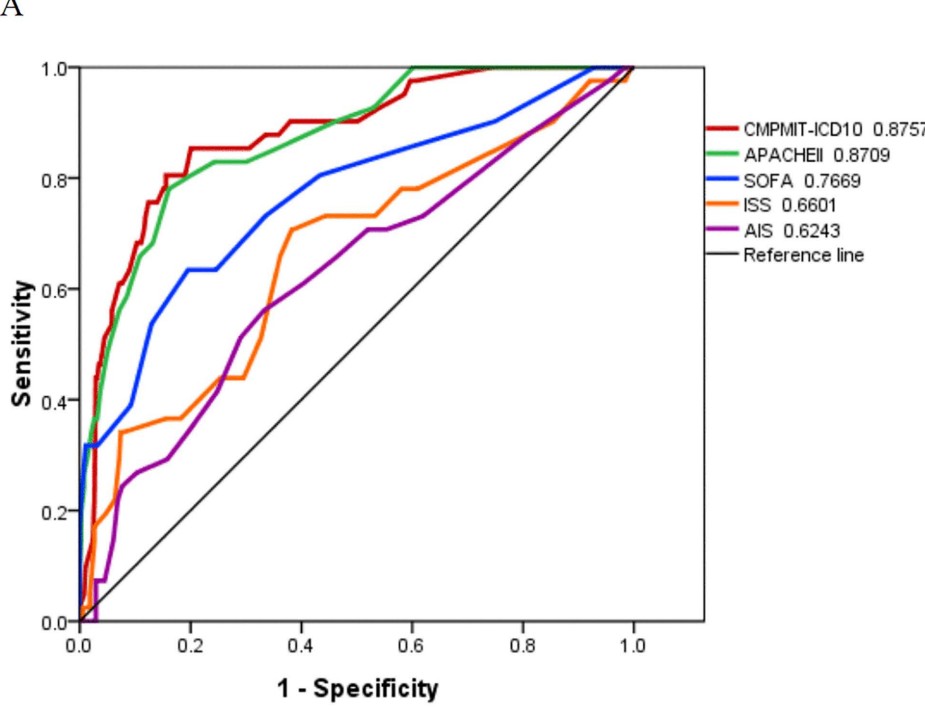

| CMPMIT-ICD10 | 0.8757 |
| APACHEII | 0.8709 |
| SOFA | 0.7669 |
| ISS | 0.6601 |
| AIS | 0.6243 |
| Reference line | |

B  Comparisons of AUCs

| CMPMIT_ICD10 | | | | |
|---|---|---|---|---|
| P = 0.8889 | APACHE II | | | |
| P = 0.0221 | P = 0.0038 | SOFA | | |
| P = 0.0001 | P < 0.0001 | P = 0.0032 | ISS | |
| P < 0.0001 | P < 0.0001 | P=0.0007 | P=0.1209 | AIS |

**Fig 2. Comparisons of areas under the receiver operating characteristic curves. A** ROC curves for predicting in-hospital mortality. **B** Pairwise comparisons of the AUCs among the five scores.

developed a CMPMIT-ICD-10 score specifically tailored for Chinese trauma patients. Therefore, in this study, we chose the CMPMIT-ICD-10 score and compared it with four other scores for the prediction of in-hospital death in THS patients.

## ISS and SOFA had limited prognostic value in THS patients

The ISS is the preferred method for providing an overall score for patients with multiple injuries. Meredith et al. [29] conducted a retrospective analysis of data from the National Trauma Data Bank (NTDB) of the USA and revealed that the ISS performed well in predicting in-hospital death in trauma patients, with an average AUC of 0.876. Additionally, another large-scale retrospective study [30] in New Zealand revealed that the ISS had a high predictive value (AUC 0.847) for 60-day mortality due to trauma. However, Rutledge et al. [31] analyzed trauma data from the North Carolina Trauma Registry (NCTR)

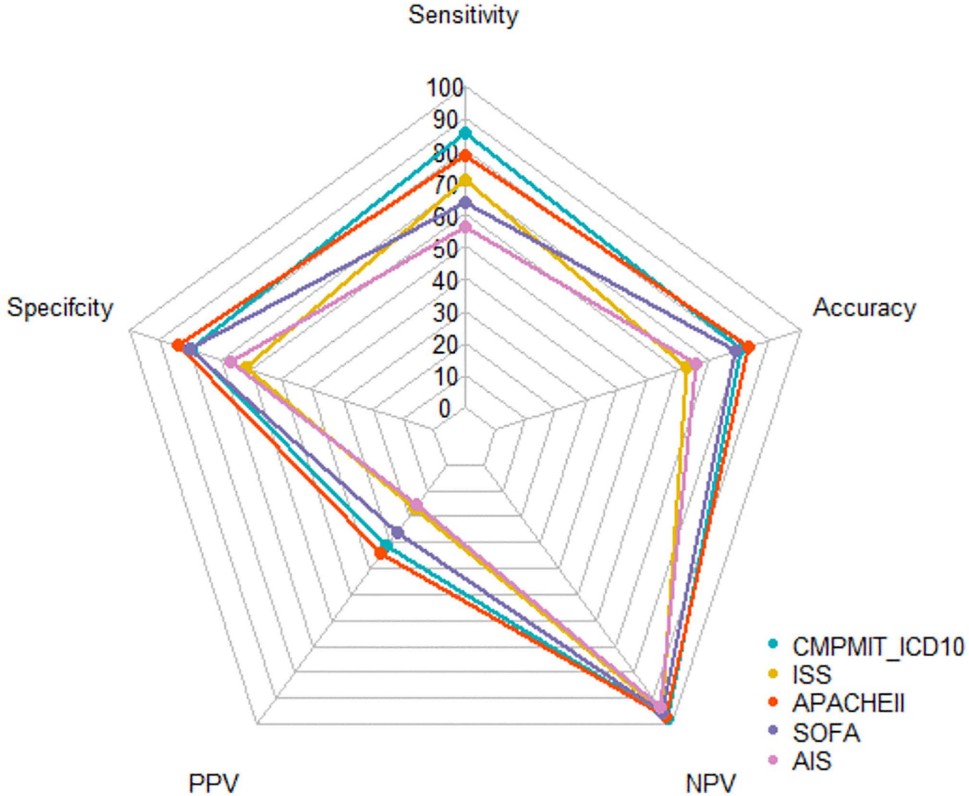

**Fig 3. Sensitivity, specificity, PPV, NPV, and accuracy of the five scores for in-hospital mortality at their respective best cutoff values.**

and reported that the ISS had limited predictive value for the prognosis of trauma patients, with an AUC of only 0.667, which was consistent with our findings (AUC of 0.660). Another study [32] revealed that for patients who were admitted to the ICU for longer than 24 hours, the ISS performed poorly in predicting in-hospital mortality (AUC 0.61). One possible reason for the differences in the AUC among different studies is the variation in patient selection. Additionally, our study focused on patients with THS who were admitted to the ICU and who typically have severe physiological disturbances, thereby limiting the predictive value of the ISS, which focuses only on anatomical injuries. Thomas et al. [33] reported that for patients with an ISS ≥ 16 without physiological risk factors, the mortality rate was very low, at 3.1%. However, with an increasing number of physiological factors, the mortality increased almost linearly, reaching 86%. Thus, incorporating physiological issues to sharpen an anatomically defined principle would help to identify critical trauma patients.

A multicenter study [34] showed that the SOFA score can reliably reflect organ dysfunction/failure in trauma patients. A cohort study conducted in Brazil on critically ill trauma patients [35] revealed that the SOFA score within 24 hours of admission performed well in predicting in-hospital mortality (AUC 0.807), significantly outperforming the ISS (AUC 0.616). These findings were consistent with the results of our study (AUC 0.767). Another prospective study of trauma patients in the ICU in Taiwan [36] revealed that the SOFA score had relatively poor predictive value for in-hospital mortality (AUC 0.707). Therefore, the SOFA score has limited value in predicting in-hospital mortality in trauma patients.

## The APACHE II score has good prognostic value in THS patients

The APACHE II score is widely used to assess the severity of disease in critically ill patients. However, the APACHE II score does not reflect anatomical injuries; thus, it cannot fully represent the severity of injury. Therefore, some researchers

**Table 4. Logistic regression analyses for the predictive value of the CMPMIT-ICD-10 score, APACHE II score, SOFA score, ISS, and AIS score.**

| Variables | Univariate | | Multivariate | |
|---|---|---|---|---|
| | OR (95% CI) | *p* value | OR (95% CI) | *p* value |
| All(n = 420) | | | | |
| CMPMIT-ICD-10 | 1.099(1.073–1.125) | <0.001 | 1.057(1.028–1.087) | <0.001 |
| APACHEII | 1.254(1.182–1.330) | <0.001 | 1.125(1.045–1.211) | 0.002 |
| SOFA | 1.405(1.266–1.558) | <0.001 | | |
| ISS | 1.056(1.026–1.088) | <0.001 | | |
| AIS | 1.072(1.018–1.129) | 0.009 | | |
| Mechanical ventilation(n = 325) | | | | |
| CMPMIT-ICD-10 | 1.090(1065–1.116) | <0.001 | 1.050(1.021–1.080) | <0.001 |
| APACHEII | 1.238(1.167–1.313) | <0.001 | 1.134(1.050–1.223) | 0.001 |
| SOFA | 1.344(1.208–1.494) | <0.001 | | |
| ISS | 1.037(1.005–1.070) | 0.024 | | |
| AIS | 1.033(0.976–1.093) | 0.258 | | |
| MODS(n = 150) | | | | |
| CMPMIT-ICD-10 | 1.101(1.065–1.137) | <0.001 | 1.060(1.021–1.100) | 0.002 |
| APACHEII | 1.287(1.176–1.409) | <0.001 | 1.161(1.049–1.286) | 0.004 |
| SOFA | 1.473(1.245–1.742) | <0.001 | | |
| ISS | 1.047(1.008–1.088) | 0.018 | | |
| AIS | 1.029(0.962–1.102) | 0.401 | | |
| Non-MODS(n = 270) | | | | |
| CMPMIT-ICD-10 | 1.084(1.045–1.124) | <0.001 | 1.099(1.039–1.162) | <0.001 |
| APACHEII | 1.174(1.071–1.286) | <0.001 | | |
| SOFA | 1.155(0.929–1.435) | 0.194 | | |
| ISS | 0.984(0.913–1.060) | 0.667 | 0.807(0.678–0.960) | 0.015 |
| AIS | 1.028(0.916–1.154) | 0.635 | | |
| ICU LOS ≥ 7d(n = 224) | | | | |
| CMPMIT-ICD-10 | 1.073(1.043–1.103) | <0.001 | 1.040(1.00–21.078) | 0.036 |
| APACHEII | 1.190(1.109–1.278) | <0.001 | 1.138(1.026–1.261) | 0.014 |
| SOFA | 1.118(0.962–1.298) | 0.145 | | |
| ISS | 0.984(0.937–1.034) | 0.529 | | |
| AIS | 0.913(0.828–1.007) | 0.069 | | |
| ICU LOS < 7d(n = 196) | | | | |
| CMPMIT-ICD-10 | 1.154(1.101–1.208) | <0.001 | | |
| APACHEII | 1.404(1.238–1.592) | <0.001 | 1.278(1.056–1.547) | 0.012 |
| SOFA | 1.967(1.540–2.512) | <0.001 | 1.505(1.046–2.165) | 0.028 |
| ISS | 1.188(1.119–1.261) | <0.001 | | |
| AIS | 1.449(1.274–1.648) | <0.001 | | |

*CMPMIT-ICD-10,* China Mortality Prediction Model in Trauma Based on the ICD-10-CM lexicon; *APACHE*II, Acute Physiology and Chronic Health Evaluation II; *SOFA,* Sequential Organ Failure Assessment; *ISS,* Injury Severity Score; *AIS,* Abbreviated Injury Scale; *MODS,* Multiple Organ Dysfunction Syndrome; *LOS,* Length of Stay; *ICU,* Intensive Care Unit

[37–38] have reported that the APACHE II score is not effective at predicting outcomes for ICU trauma patients. However, subsequent studies [39–40] reported that the APACHE II score was a good predictor of mortality in ICU trauma patients. A prospective study by Wong et al. [41] revealed that the APACHE II score had a greater ability to predict mortality in 470

ICU trauma patients in Canada (AUC 0.92±0.02). A 4-year retrospective study in South Korea involving 706 ICU trauma patients by Hwang et al. [42] reached comparable conclusions (AUC 0.950). These findings were consistent with our observations, where the APACHE II score demonstrated significantly better predictive value for prognosis (AUC 0.875) compared with the ISS (AUC 0.660). Therefore, the APACHE II score performed well in predicting in-hospital death in THS patients.

**The ISS, SOFA, and APACHE II scores have low sensitivity, good specificity and good accuracy in predicting mortality in THS patients**

Fueglistaler et al. [35] reported that the ISS had a low sensitivity (38.9%)for the prediction of 30-day mortality but a high specificity (96.2%) and accuracy (83.1%). A meta-analysis conducted by Deng et al. [43], which included 11 articles evaluating the performance of the ISS in predicting mortality, demonstrated that the sensitivity and specificity of the ISS varied widely across studies. This meta-analysis revealed a sensitivity of 0.64 and a specificity of 0.93 for the ISS. The sensitivity was lower than that of our study, but the specificity was greater. The possible reasons for these differences included differences in study quality, sample size, and economic characteristics across countries.

Additionally, a study by Hwang et al. [42] revealed that for ICU trauma patients, the sensitivity, specificity, and accuracy of the SOFA score were 74.1%, 97.1%, and 92.4%, respectively, and these values were greater than our results (63.4%, 80.5%, and 78.8%, respectively). However, another single-center study from Switzerland [35] revealed that the SOFA score had a sensitivity of only 20.4% for predicting 30-day mortality, while the specificity was relatively high at 94.0%, and the accuracy was slightly lower at 77.2%. The main reason for these differences might be related to the variation in cutoff values chosen in different studies. Interestingly, two different studies from different times and regions revealed that the APACHEII score has remarkably similar sensitivity, specificity, and accuracy—50.8%, 97.3%, and 91.1%, respectively—in a Canadian study [41] and 58.5%, 99.6%, and 91.1%, respectively, in a South Korean study [42]. In these two studies, the sensitivity of the APACHE II score was slightly lower than that in our study, but the specificity and accuracy were greater. This difference is likely related to the selection of criteria, as both studies analyzed the ability to predict group mortality by the probability of death (>0.5) calculated for each patient on the basis of the APACHE II score rather than the specific value. Therefore, the criteria were stricter. Although this scoring system effectively identified surviving patients and reduced unnecessary treatment and resource waste, the low sensitivity means that some patients who were at risk of dying were not identified promptly, potentially missing opportunities for intervention.

**Advantages of CMPMIT-ICD-10 and its good performance in predicting mortality in THS patients**

The establishment of trauma centers in China is still in its infancy, with the mortality and disability rates of severe trauma patients remaining higher than those in developed countries [44]. Currently, a proper method for evaluating the quality of care at these trauma centers is not available. The CMPMIT-ICD-10 was established on the basis of a trauma database in China and is important for strengthening trauma care capacities in China [8]. The CMPMIT-ICD-10 adequately considered the influence of comorbidities and posttraumatic physiological responses on mortality. Another notable advantage is that mortality prediction only requires baseline information such as age, sex, comorbidities, ICD-10-CM codes, and consciousness status, which may be more convenient for clinical application. The internal validation results of the CMPMIT-ICD-10 were good, with strong discrimination and calibration; as an external validation of the CMPMIT-ICD-10, this study further confirmed its robustness.

These results indicated that the CMPMIT-ICD-10 has advantages in specific groups while reflecting the severity of their condition and risk of mortality more accurately. Although the CMPMIT-ICD-10 score had a high AUC (0.9195) and sensitivity (90.0%) in patients with shorter hospital stays (<7 days), we also found that it may be limited in predicting in-hospital mortality in these patients, which requires further research and validation. Subgroup analysis further confirmed the robustness and reliability of the CMPMIT-ICD-10 in predicting in-hospital mortality among patients with THS. These

 

findings are important for clinical decision-making, providing more specific guidance for clinicians to make more accurate prognostic assessments and treatment decisions on the basis of the particular situations of patients, thus improving patient outcomes.

## Study limitations

Despite the great predictive performance of the CMPMIT-ICD-10 demonstrated by this study, several limitations should be considered. First, this study is limited by its retrospective design, and all of the patients were from the same hospital. We attempted to minimize this effect by consecutively recruiting patients and adjusting for confounding factors using a multivariate regression model. Second, although the CMPMIT-ICD-10 accounts for variables such as age, underlying illnesses, and post trauma physiological responses, it may not completely eliminate the impacts of the coding rules because of the possibility of not accurately quantifying the severity of injuries.Moreover, although sensitivity analysis and subgroup analysis were conducted to further confirm our findings, the impact of a relatively limited sample size on the results cannot be ruled out. In the future, we will conduct multicenter studies to expand the sample size and compare our results with those of other scoring systems, such as the Trauma and Injury Severity Score (TRISS) and TMPM-ICD-10, to enhance the reliability and applicability of our scoring system in clinical practice.

## Conclusions

The CMPMIT-ICD10 performed well in predicting short-term mortality in patients with THS in the ICU, was comparable to the APACHE II score, and was significantly better than the SOFA score, ISS, and AIS. These findings suggest that the CMPMIT-ICD-10 may have superior utility for predicting death in THS patients in an ICU setting.

## Supporting information

**S1 File. S1-8 Tables and S1-6 Figures.**
(DOCX)

## Acknowledgments

Not applicable.

## Author contributions

**Conceptualization:** haiyan xue, shu li, zhenzhou wang, tianbing wang, Xiujuan ZHAO.

**Data curation:** jie cai, feifei jin, shu li, chun fu.

**Formal analysis:** jie cai, gang zhou, feifei jin, haiyan xue, Xiujuan ZHAO.

**Funding acquisition:** fengxue zhu.

**Investigation:** jie cai, gang zhou, haiyan xue, zhenzhou wang, wei huang, Xiujuan ZHAO.

**Methodology:** jie cai, gang zhou, feifei jin, chun fu, Xiujuan ZHAO.

**Project administration:** shu li, zhenzhou wang, wei huang, tianbing wang.

**Resources:** gang zhou, haiyan xue, shu li.

**Software:** gang zhou, feifei jin, haiyan xue, shu li, zhenzhou wang.

**Supervision:** gang zhou, chun fu, zhenzhou wang, wei huang, tianbing wang, fengxue zhu.

**Validation:** jie cai, feifei jin, tianbing wang, fengxue zhu, Xiujuan ZHAO.

**Visualization:** chun fu.

**Writing – original draft:** jie cai, gang zhou, feifei jin.

**Writing – review & editing:** wei huang, tianbing wang, fengxue zhu, Xiujuan ZHAO.

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
