## [Decision Letter · Decision Letter 0]

14 Oct 2025

Dear Dr. ZHAO,

Thank you for submitting your manuscript to PLOS ONE. After careful consideration, we feel that it has merit but does not fully meet PLOS ONE’s publication criteria as it currently stands. Therefore, we invite you to submit a revised version of the manuscript that addresses the points raised during the review process.

We look forward to receiving your revised manuscript.

Kind regards,

Alaa Oteir, PhD

Academic Editor

PLOS ONE

Journal Requirements:

3. Thank you for stating the following in your manuscript:

[This study was supported by the National Natural Science Foundation of China (81971808), the Beijing Natural Science Foundation (7222199), Capital Health Research and Development of Special (2018-2-4082) and the Peking University People’s Hospital Scientific Research Development Funds (RDGS2023-07). The funders played no role in the design or implementation of the study, or manuscript writing.]

[The author(s) received no specific funding for this work.]

6. Please include captions for your Supporting Information files at the end of your manuscript, and update any in-text citations to match accordingly. Please see our Supporting Information guidelines for more information: http://journals.plos.org/plosone/s/supporting-information .

Reviewers' comments:

Reviewer's Responses to Questions

**Comments to the Author**

1. Is the manuscript technically sound, and do the data support the conclusions?

Reviewer #1: Yes

Reviewer #2: Yes

2. Has the statistical analysis been performed appropriately and rigorously?

Reviewer #1: Yes

Reviewer #2: Yes

3. Have the authors made all data underlying the findings in their manuscript fully available?

Reviewer #1: Yes

Reviewer #2: Yes

4. Is the manuscript presented in an intelligible fashion and written in standard English?

Reviewer #1: Yes

Reviewer #2: Yes

Reviewer #1: Thank you for inviting me to review the article entitled ‘Prognostic Accuracy of the CMPMIT-ICD10, APACHE-II, SOFA, ISS, and AIS Scoring Scale for In-Hospital Death Among Patients with Traumatic Haemorrhagic Shock’.

Serious injuries are the leading cause of death and disability worldwide. Unintentional injuries are ranked as the sixth leading cause of death and the fifth leading cause of moderate to severe disability. The objective of the study was to compare the prognostic performance of the CMPMIT-ICD10 with that of the Acute Physiology and Chronic Health Evaluation II, Sequential Organ Failure Assessment, Injury Severity Score, and Abbreviated Injury Scale for in-hospital mortality in patients with traumatic haemorrhagic shock(THS).

Title:

• The title of the paper reflect the scope of the research.

Abstract:

• The abstract clearly describes the work.

Introduction:

• This section provides a good introduction to the reader's topic.

Methods:

• In the methods section, it is worth adding a description of the elements/parameters of each scale and the range of possible scores on these scales. This can even take the form of a table. This will make it easier for the reader to understand the results of the work.

Results:

• This section is well described.

Discussion:

• Dividing this into subsections, where the results are discussed according to individual scales, would help the reader to understand the topic.

References:

• The references are correctly selected.

I congratulate the authors of the study. The article is very well prepared and clinically useful.

Reviewer #2: Abstract:

Switch out “the good specificity” for “a good specificity.” Make it crystal clear right off the bat: spell out the study period (2013–2023) and describe the study design upfront.

Introduction:

Some of those stats—like the mortality rates from haemorrhagic shock—could really use extra references to back them up. like:

*Ala A, Shams Vahdati S, Asghari A, Makouei M, Poureskandari M. Accuracy of the new injury severity score in the evaluation of patients with blunt trauma. Archives of Trauma Research. 2022 Jun 1;11(2):71-3.

*Hakimzadeh Z, Vahdati SS, Ala A, Rahmani F, Ghafouri RR, Jaberinezhad M. The predictive value of the Kampala Trauma Score (KTS) in the outcome of multi-traumatic patients compared to the estimated Injury Severity Score (eISS). BMC emergency medicine. 2024 May 14;24(1):82.

*

Methods:

Actually list the ICD-10 codes or categories you used for the CMPMIT scoring, instead of leaving readers guessing. Toss in a brief bit about how you made sure the Trauma-Specific Database data was accurate—something on data validation and quality checks.

Tables & Figures:

Table 1: Keep the decimal points consistent—so, for example, use <0.001 instead of 0.001.

Table 2: Make sure to clearly label the units for AUC, PPV, and NPV. Use bold to highlight the best-performing numbers.

Figure 2: The ROC curve comparison is kinda hard to read—add color coding and a clear legend, maybe even replot it so differences jump out.

Explicitly mention Supplementary Tables S1–S7 in the main text so readers don’t miss them.

Language & Style:

The writing’s already pretty clear, just needs a few tweaks for grammar. For example:

Instead of: “The CMPMIT-ICD10 had the highest sensitivity (85.4%), the good specificity (79.9%)…”

Go with: “The CMPMIT-ICD10 had the highest sensitivity (85.4%), good specificity (79.9%), and overall accuracy (80.4%).”

Ditch the repetition—no need to say “good specificity and good accuracy” back-to-back.

Ethics & Funding:

Ethics approval (2020PHB258-01) and the waiver of consent are both there and easy to find—nice job.

Double-check that the Funding and Competing Interests sections use the latest PLOS ONE format. Don’t want any surprises there.

**Do you want your identity to be public for this peer review?** For information about this choice, including consent withdrawal, please see our Privacy Policy

Reviewer #1: No

Reviewer #2: No

---

## [Author Response · Author response to Decision Letter 1]

7 Nov 2025

Dear Editor Alaa Oteir,

We are very grateful for your and reviewers' critical comments and thoughtful suggestions on our manscript, we have made careful modification on the original manuscript. We acknowledge your comments and constructive suggestions very much, which are very valuable in improving the quality of our manuscript. Here are our responses to the reviewers' comments.

Reviewer #1:

Methods:

• In the methods section, it is worth adding a description of the elements/parameters of each scale and the range of possible scores on these scales. This can even take the form of a table. This will make it easier for the reader to understand the results of the work.

Response: Thank you very much for your advice. We have modified it according to your requirements. We have added Table1 for the scores of CMPMIT-ICD10.

Discussion:

• Dividing this into subsections, where the results are discussed according to individual scales, would help the reader to understand the topic.

Response: Thanks for your suggestion. We have divided the Discussion into subsections according to your requirements and added sub-headings.

Reviewer #2:

Abstract:

Switch out “the good specificity” for “a good specificity.” Make it crystal clear right off the bat: spell out the study period (2013–2023) and describe the study design upfront.

Response: Thank you very much for your advice. We have modified it according to your requirements. We have revise it as "This retrospective observational cohort study was conducted at a tertiary teaching hospital from May 1, 2013 to May 31, 2023." in the Abstract.

Introduction:

Some of those stats—like the mortality rates from haemorrhagic shock—could really use extra references to back them up. like:

*Ala A, Shams Vahdati S, Asghari A, Makouei M, Poureskandari M. Accuracy of the new injury severity score in the evaluation of patients with blunt trauma. Archives of Trauma Research. 2022 Jun 1;11(2):71-3.

*Hakimzadeh Z, Vahdati SS, Ala A, Rahmani F, Ghafouri RR, Jaberinezhad M. The predictive value of the Kampala Trauma Score (KTS) in the outcome of multi-traumatic patients compared to the estimated Injury Severity Score (eISS). BMC emergency medicine. 2024 May 14;24(1):82.

*

Response: Thanks for your suggestion. We have modified it according to your requirements. We have revise it as " haemorrhagic shock related to trauma is the main cause of early in-hospital deaths, accounting for 17.1% to 50% of all trauma-related deaths[3-5]" and add the 2 references.

Methods:

Actually list the ICD-10 codes or categories you used for the CMPMIT scoring, instead of leaving readers guessing. Toss in a brief bit about how you made sure the Trauma-Specific Database data was accurate—something on data validation and quality checks.

Response: Thank you very much for your advice. We have modified it according to your requirements. We have added Table S1 for the ICD-10 code of trauma areas and complications for CMPMIT-ICD10. We have add "To ensure the highest data integrity and accuracy, we implemented a multi-layered validation process. This included automated validation rules, logic and consistency checks, and regular audit and reconciliation. This rigorous approach was crucial for maintaining the database's reliability and accuracy for both clinical decision-making and research analysis." in the Data collection.

Tables & Figures:

Table 1: Keep the decimal points consistent—so, for example, use <0.001 instead of 0.001.

Table 2: Make sure to clearly label the units for AUC, PPV, and NPV. Use bold to highlight the best-performing numbers.

Figure 2: The ROC curve comparison is kinda hard to read—add color coding and a clear legend, maybe even replot it so differences jump out.

Explicitly mention Supplementary Tables S1–S7 in the main text so readers don’t miss them.

Response: Thanks for your suggestion. We have modified it according to your requirements. We checked the decimal points consistent again in Table 1. We have labeled the units for AUC, PPV, and NPV, and bold to highlight the best-performing numbers in Table 2. We have redrawed the ROC curve and added color coding and a clear legend.

Language & Style:

The writing’s already pretty clear, just needs a few tweaks for grammar. For example:

Instead of: “The CMPMIT-ICD10 had the highest sensitivity (85.4%), the good specificity (79.9%)…”

Go with: “The CMPMIT-ICD10 had the highest sensitivity (85.4%), good specificity (79.9%), and overall accuracy (80.4%).”

Ditch the repetition—no need to say “good specificity and good accuracy” back-to-back.

Response: Thank you very much for your advice. We have modified it according to your requirements. And once again invited local English experts to polish the whole manuscript.

Ethics & Funding:

Ethics approval (2020PHB258-01) and the waiver of consent are both there and easy to find—nice job.

Double-check that the Funding and Competing Interests sections use the latest PLOS ONE format. Don’t want any surprises there.

Response: Thanks for your suggestion. We have checked the Funding and Competing Interests section.

---

## [Decision Letter · Decision Letter 1]

17 Dec 2025

Prognostic accuracy of the CMPMIT-ICD-10, APACHE Ⅱ, SOFA, ISS, and AIS for in-hospital death among patients with traumatic hemorrhagic shock

PONE-D-24-58143R1

Dear Dr. ZHAO,

We’re pleased to inform you that your manuscript has been judged scientifically suitable for publication and will be formally accepted for publication once it meets all outstanding technical requirements.

Kind regards,

Alaa Oteir, PhD

Academic Editor

PLOS One

---

## [Editor Report · Acceptance letter]

PONE-D-24-58143R1

PLOS One

Dear Dr. ZHAO,

I'm pleased to inform you that your manuscript has been deemed suitable for publication in PLOS One. Congratulations! Your manuscript is now being handed over to our production team.

Kind regards,

on behalf of

Dr. Alaa Oteir

Academic Editor

PLOS One